# Stratifying Antenatal Hydronephrosis: Predicting High-Grade VUR Using Ultrasound and Scintigraphy

**DOI:** 10.3390/diagnostics14040384

**Published:** 2024-02-09

**Authors:** Niklas Pakkasjärvi, Sofia Belov, Timo Jahnukainen, Reetta Kivisaari, Seppo Taskinen

**Affiliations:** 1Department of Pediatric Surgery, Helsinki University Hospital, 000290 Helsinki, Finland; sofia.belov@hus.fi (S.B.); seppo.taskinen@hus.fi (S.T.); 2Department of Pediatric Nephrology and Transplantation, Helsinki University Hospital, University of Helsinki, 000290 Helsinki, Finland; timo.jahnukainen@hus.fi; 3Department of Pediatric Radiology, Helsinki University Hospital, 000290 Helsinki, Finland; reetta.kivisaari@hus.fi

**Keywords:** antenatal hydronephrosis, scintigraphy, vesicoureteral reflux, risk stratification, diagnostic process

## Abstract

(1) Background: Antenatal hydronephrosis (AHN), detected in approximately one percent of prenatal ultrasounds, is caused by vesicoureteral reflux (VUR) in 15–21% of cases, a condition with significant risks such as urinary tract infections and renal scarring. Our study addresses the diagnostic challenges of VUR in AHN. Utilizing renal ultrasonography and scintigraphy, we developed a novel scoring system that accurately predicts high-grade VUR, optimizing diagnostic precision while minimizing the need for more invasive methods like voiding cystourethrogram (VCUG); (2) Methods: This retrospective study re-analyzed renal ultrasonography, scintigraphy, and VCUG images from infants admitted between 2003 and 2013, excluding cases with complex urinary anomalies; (3) Results: Our analysis included 124 patients (75% male), of whom 11% had high-grade VUR. The multivariate analysis identified visible ureter, reduced renal length, and decreased differential renal function (DRF) as primary predictors. Consequently, we established a three-tier risk score, classifying patients into low, intermediate, and high-risk groups for high-grade VUR, with corresponding prevalences of 2.3%, 22.2%, and 75.0%. The scoring system demonstrated 86% sensitivity and 79% specificity; (4) Conclusions: Our scoring system, focusing on objective parameters of the visible ureter, renal length, and DRF, effectively identifies high-grade VUR in AHN patients. This method enhances diagnostics in ANH by reducing reliance on VCUG and facilitating more tailored and less invasive patient care.

## 1. Introduction

With a prevalence of about one percent, antenatal hydronephrosis remains one of the most frequently observed abnormalities seen in prenatal ultrasonography [1,2]. It can be attributed to a variety of etiologies, and importantly, only a subset of antenatal hydronephrosis cases necessitate active treatment. The risk of postnatal pathology in antenatal hydronephrosis is notably correlated with its severity, ranging from 12% in mild cases to as high as 88% in severe cases [3]. Key predictors of adverse postnatal outcomes include oligohydramnios, antenatal hydronephrosis, an enlarged bladder, and the ‘keyhole sign’ (appearance of a dilated proximal urethra in conjunction with a thick-walled, distended bladder), indicative of severe bladder outlet obstruction [4,5]. Other critical indicators include increased renal echogenicity, poor cortico-medullary differentiation, parenchymal thinning, calyceal dilation, renal cysts, and urinoma [6,7,8,9,10]. Antenatal hydronephrosis identified in the third trimester particularly warrants close postnatal monitoring through renal ultrasound.

Although obstructive pathologies in the form of ureteropelvic junction obstruction and ureterovesical junction obstruction are predominant, vesicoureteral reflux (VUR) accounts for 15–21% of cases of postnatal hydronephrosis findings [10,11]. Correct diagnosis becomes crucial, given that each condition warrants specific attention. VUR is a complex condition with variable symptoms and outcomes. To mitigate the risks of urinary tract infections, renal scarring, and reflux nephropathy associated with VUR, current treatment protocols universally advocate for individualized and risk-stratified approaches [12]. Despite the plethora of classification systems for hydronephrosis, there is an evident need for a more precise characterization of the risks associated with high-grade VUR [13]. Previous research has underscored the correlation between high-grade VUR and urinary tract infections (UTI) [14,15,16], which our preceding study aligned with, showing that only higher-grade VUR is associated with breakthrough infections [16]. However, further tools are needed for improved diagnostic workflows. It is imperative to stress the clinical significance of identifying high-grade vesicoureteral reflux (VUR) before the occurrence of breakthrough infections. These infections carry a substantial risk of causing kidney damage, making early detection and intervention crucial. By accurately diagnosing high-grade VUR at an early stage, we can significantly reduce the likelihood of renal complications and ensure better long-term renal health outcomes. This approach aligns with the overarching goal of preemptive healthcare, which focuses on preventing damage before it occurs rather than merely responding to its consequences.

Guidelines for VUR diagnostics vary regionally. Traditionally, VUR is diagnosed with a voiding cystourethrogram (VCUG); however, this study modality entails ionizing radiation and predisposes the patients to both catheterizations and UTIs [17,18,19]. Consequently, alternative strategies have been developed to identify patients at the most significant risk of high-grade VUR, aiming to reduce unwarranted VCUGs [20,21,22,23]. Observations from previous studies highlight the utility of renal ultrasonography and renal scintigraphy in predicting the risk of grade 4–5 VUR [20,21,22,23]. The inclusion of renal length in the analysis is supported by literature indicating that kidneys affected by VUR tend to be smaller. Zerin and Leiser demonstrated the impact of vesicoureteral reflux on contralateral renal length in infants with multicystic dysplastic kidneys [24]. Similarly, Roihuvuo-Leskinen et al. found an association between adult kidney size and childhood vesicoureteral reflux [25]. In terms of ultrasonography findings, SFU grades 3 and 4 are more commonly associated with high-grade VUR, as evidenced by Lee et al. in their evaluation of prenatal hydronephrosis [26]. Hansson et al. showed that DMSA scintigraphy, which detected grade ≥3 VUR with 96% sensitivity and 53% specificity, could obviate the need for VCUG in patients with a history of UTI when normal [23]. The primary objective of this study is to devise and introduce a scoring system that integrates insights from renal ultrasonography and scintigraphy to enhance the diagnostic accuracy for grade 4–5 VUR in patients with antenatal hydronephrosis. Additionally, we aim to evaluate whether this scoring system could reduce the reliance on VCUG in the diagnostic continuum. Our study is designed with a dual focus: firstly, to enhance the diagnostic process for early identification of the risk of high-grade vesicoureteral reflux, and secondly, to achieve this identification before the onset of infections. This proactive approach is crucial in safeguarding the developing kidneys from disturbances that can impede proper growth. Moreover, by accurately identifying patients at risk, we can provide timely interventions that are essential for addressing and managing pathological hydronephrosis. In doing so, our aim is not only to improve the immediate clinical outcomes but also to contribute to the long-term renal health of the affected individuals. The ultimate goal of an enhanced diagnostic capability is to ensure that interventions are implemented at the most opportune time, thus optimizing patient care and preserving kidney function in a critical phase of renal development.

## 2. Materials and Methods

This study was approved by the institutional ethics committee. All work has been done in accordance with the Helsinki Declaration. As the study involved a review of existing medical records and did not influence patient treatment or outcomes, individual patient consent was not required. Patient data were anonymized, and confidentiality was maintained by assigning unique codes to each case, following data protection and privacy guidelines.

We identified all patients who were diagnosed between 2003 and 2013 with urinary tract anomalies (ICD-10: Q60.0–Q64.9 and N13.0–N13.9) from our unit’s electronic database. Antenatal Ultrasound Protocol: Antenatal ultrasounds were mostly conducted around 18 to 21 weeks of gestation, following national guidelines. For suspected hydronephrosis cases, additional ultrasounds were performed later in pregnancy. The diagnostic criterion for inclusion was a renal pelvic AP diameter over 7 mm in the third trimester, with a confirmatory postnatal ultrasound within 2 to 7 days. Due to the retrospective nature of our study, exact ultrasound dates for each case were not available, but this consistent criterion was applied across all cases. Our focus centered on patients with persistent hydronephrosis (postnatal anteroposterior diameter of the renal pelvis >7 mm) or a visible distal ureter on renal ultrasonography, which we included in the study. Each patient underwent a comprehensive diagnostic evaluation, including renal ultrasonography, VCUG and renal scintigraphy (Dimercaptosuccinic acid (DMSA), Mercaptoacetyltriglycine (MAG3) or Diethylenetriaminepenta-acetate (DTPA). Of 192 neonates initially evaluated, 124 (65%) were included in the study. The remaining 68 neonates were excluded from the study for the following reasons: (1) Lack of sufficient imaging data, (2) Postnatally detected hydronephrosis, (3) Presence of complex urinary tract anomalies, including but not limited to ureterocele, duplex collecting system, and bladder outlet obstructions, (4) Syndromic patients, (5) Dysplasia, as determined based on visual criteria, such as the presence of multiple cysts or the absence of corticomedullary differentiation in non-hydronephrotic, high-echogenic kidneys.

According to the protocol, VCUG was performed at 4 weeks of age if the patient had the following: (1) A visible ureter on renal ultrasonography, (2) Renal pelvic anteroposterior diameter ≥ 10 mm, (3) hydronephrosis and reduced kidney size, (4) bilateral hydronephrosis. VUR grading adhered to the international classification system [27]. Based on insights gained from our previous investigations [9], only patients with grade 4–5 VUR had a significantly increased risk of UTI, prompting further subdivision of the VUR patients into two subgroups, grade 1–3 VUR and grade 4–5 VUR.

Ultrasonographies were performed in accordance with standard practices, evaluating the following: urinary tract anatomy, bladder wall, presence of ureterocele, renal length, echogenicity of the renal cortex and medulla, possible scars, other lesions or calculi, renal arterial and venous blood flow, anteroposterior diameter of the renal pelvis, the shape of calyces, and the diameter of distal and proximal visible ureters. All images were re-evaluated by an experienced pediatric radiologist (RK). Comprehensive data, including kidney length, renal pelvic APD, Society of Fetal Urology (SFU) grade of hydronephrosis [28], distal ureter diameter (if visible), and renal parenchyma, were extracted from the RBUS images. The renal parenchyma, evaluated from the midcoronal section of the kidneys, was classified as either normal or reduced. In final scoring, the ureter was classified as visible or invisible, and renal length was categorized into three groups: (1) >53 mm (>0 SD), (2) 46–53 mm (0–(−1) SD), and (3) <46 mm (<(−1) SD). The length of the kidney was not available in one renal unit (RU).

In instances of obstructive uropathy suspicion or enlarged hydronephrotic kidney, MAG3/DTPA scintigraphy was employed, while DMSA scintigraphy was used for the assessment of small-sized kidneys with minor (SFU grade ≤ 2) hydronephrosis. In the final scoring, differential renal function (DRF) was divided into three categories: (1) >50%, (2) 44–50%, (3) <44%. MAG3/DTPA scintigraphy was performed on 11 patients (9%), DMSA scintigraphy on 17 (14%), and both DMSA and MAG3/DTPA scintigraphy on 96 (77%) patients. The risk of grade 4–5 VUR was assessed in relation to kidney length, anteroposterior diameter (APD) of the renal pelvis, SFU grading of hydronephrosis and the differential renal function (DRF). Scoring is presented in the table.

Information regarding each patient’s age and gender and comprehensive results from all relevant imaging studies were systematically collected from patient records. This included a detailed analysis of ultrasonography, renal scintigraphy, and other pertinent diagnostic imaging, providing a foundation for the data set.

### Statistical Analysis

Statistical analyses were conducted using the R software (R package version 3.3.3). Univariate and multivariate logistic regression analyses were employed to investigate the association between grade 4–5 vesicoureteral reflux (VUR) and key variables. Receiver operating characteristic (ROC) curve analysis was used to assess the accuracy of the predictive model, with the area under the curve (AUC) serving as a measure of model performance. It is important to note that, given the complexity of the data, a specific correction method for multiple comparisons was not applied in this analysis.

## 3. Results

### 3.1. General Findings

We included a total of 124 patients (247 RUs) during the study period of 2003–2013, the majority of whom were males (75%; Table 1). Of the RUs, 48 (19%) exhibited any grade VUR, and 28 (11%) RUs demonstrated grade 4–5 VUR. The median age at the renal ultrasonography examination was 1.3 (range 0.1–3.0) months, while the median age at the first renal scintigraphy (DMSA, MAG3 or DTPA) was 1.4 (range 0.8–15.6) months.

### 3.2. Parameters That Predict VUR

To discern the risk of grade 4–5 VUR, we employed univariate analysis, considering the following parameters in renal ultrasonography: renal length, renal pelvic APD, SFU grade 0–2 versus grade 3–4, visibility of the distal ureter, and reduction of renal parenchyma and DRF in renal scintigraphy (Figure 1).

The univariate analysis pinpointed significant risk factors for grade 4–5 VUR: a visible distal ureter (OR 20.20; CI 8.43–51.49, *p* < 0.001), small renal length (OR 0.35; CI 0.18–0.64, *p* = 0.001), and SFU grade 3–4 (OR 1.68; CI 1.15–2.53, *p* = 0.010) in renal ultrasonography. Furthermore, reduced DRF in renal scintigraphy was also indicative of grade 4–5 VUR (OR 0.944; CI 0.916–0.971, *p* < 0.001). No significant correlation was found between renal length and DRF (0.1 (−0.03–0.22), *p* = 0.120).

We observed that shorter renal length, as depicted in Figure 1, was associated with an increased risk of grade 4–5 VUR. The odds ratio (OR) of 0.35 (CI 0.18–0.64, *p* = 0.001) indicated that smaller kidneys were more likely to be associated with high-grade VUR. This relationship is illustrated in Figure 1, which highlights the increasing risk of grade 4–5 VUR as renal length decreases.

Additionally, the anteroposterior diameter of the renal pelvis (APD) and SFU grading showed significant associations with VUR risk. An elevated APD and higher SFU grades were linked to an increased likelihood of high-grade VUR. These findings underscore the importance of assessing these parameters in predicting VUR severity.

Furthermore, DRF played a crucial role in our analysis. Figure 1 illustrates that reduced DRF was a significant indicator of grade 4–5 VUR, with an odds ratio of 0.944 (CI 0.916–0.971, *p* < 0.001). Patients with lower DRF values were more likely to have high-grade VUR, emphasizing the significance of assessing renal function in risk assessment.

Subsequently, a multivariate analysis was also applied to the parameters significantly associated with an increased risk of grade 4–5 VUR. For analytical clarity, renal length was divided into three categories: (1) >53 mm (>0 SD), (2) 46–53 mm (0–(−1) SD), and (3) <46 mm (<(−1) SD). DRF was similarly divided into three categories: (1) >50%, (2) 44–50% (3) <44%. A visible ureter and reduced renal length in renal ultrasonography and reduced DRF in renal scintigraphy were significantly predictive of grade 4–5 VUR (OR 19.26; CI 7.10–57.59, *p* < 0.001), (OR 2.14; CI 1.11–4.23, *p* = 0.024) and (OR 2.84; CI 1.50–5.66, *p* = 0.002). However, the SFU grade did not reach statistical significance in predicting grade 4–5 VUR (*p* = 0.052) and was thus omitted from further analysis.

#### Scoring

Building on the multivariate model, we devised a scoring system (Table 2) to predict the risk of vesicoureteral reflux, adopting a methodological framework from the study by Han et al. [29]. Central to this scoring system is the incorporation of objective key parameters: differential renal function, renal length, and the binary assessment of ureter visibility (yes/no). Patients were then stratified into three distinct risk categories: (1) Low, (2) Intermediate, (3) High. The scoring system provides a cumulative risk assessment, enabling clinicians to stratify patients according to their likelihood of developing high-grade VUR and tailor diagnostic and treatment approaches accordingly. The prevalence of grade 4–5 VUR within these categories was revealing a modest 2.3% in the low-risk group, a notable 22.2% in the intermediate group, and a significant 75.0% in the high-risk group, as depicted in Figure 1.

Figure 1 illustrates the distribution of patients across the risk categories in our scoring system. Patients in the high-risk category exhibited a substantially higher prevalence of grade 4–5 VUR (75.0%), highlighting the system’s ability to stratify patients effectively based on their risk of high-grade VUR. This stratification provides valuable clinical insights for early intervention and management.

Further analytical rigor was applied through the use of receiver operating characteristic (ROC) curve analysis, which evaluated the true positive rate (sensitivity) and false positive rate (1-specificity) of our risk scoring system in identifying renal units (RUs) with grade 4–5 VUR. The model’s area under the curve (AUC) was a robust 0.89 (95% CI 0.81–0.96), as illustrated in Figure 2. This was indicative of the system’s efficacy, with the sensitivity and specificity for detecting grade 4–5 VUR being 86% and 79%, respectively.

In addition to these findings, we also examined the outcomes of the voiding cystourethrogram results, which were classified into two categories: grade 4–5 and grade 0–3 VUR, according to earlier studies. The ROC analysis further substantiated the validity of our model. When both DRF and renal ultrasonography findings were included, the AUC remained consistent at 0.89 (95% CI 0.81–0.96, as shown in Figure 2). Interestingly, in the model variant excluding DRF, there was a slight decrease in predictive accuracy, with the AUC at 0.85 (95% CI 0.76–0.94), underscoring the importance of DRF in our risk assessment framework.

## 4. Discussion

In this study, we introduce a novel three-grade risk classification score that leverages objective measurements from ureteral visibility, kidney length, and differential renal function to predict high-grade vesicoureteral reflux with both high sensitivity and specificity. Our findings highlight the role of dilating VUR as a major risk factor for urinary tract infections, consistent with the evidence from previous studies, all of which point to a substantial association between UTIs and grade 4–5 VUR in patients with antenatal hydronephrosis [15,16].

As the standard method for detecting VUR, VCUG is one of the most common fluoroscopic examinations in children. VCUG is known to be associated with a 2–6% risk for UTI, and it exposes children to radiation [16,17,18,19,30]. Given that the clinical significance of low-grade VUR appears limited, our risk classification system offers an objective method to identify those patients who would benefit most from VCUG, particularly those at higher risk for grade 4–5 VUR [14,15,16]. By refining the risk stratification process for high-grade VUR, our study contributes to the ongoing effort to improve antenatal hydronephrosis diagnostics. This ensures more judicious use of VCUG, paving the way for individualized patient care.

Abnormal kidney length, hydroureter, and uroepithelial thickening (hypoechoic rim within the renal pelvic wall) in renal ultrasonography have all been linked to high-grade VUR [20,31]. However, renal ultrasonography is a relatively insensitive method for screening for VUR; more recently, the possibility of using renal scintigraphy in VUR diagnostics has raised interest [26,32]. A top-down approach has been an accepted method for screening VUR in patients with febrile UTIs since 2004. With this method, VCUG is only performed on patients with abnormal findings with DMSA scintigraphy [23]. In the study by Hansson et al., about 80% of the children with grade 3–5 VUR had abnormal DMSA scintigraphies. In their study, DMSA scintigraphy detected grade ≥3 VUR with 96% sensitivity and 53% specificity. The authors concluded that in patients with a history of UTI, a normal DMSA scan makes VCUG unnecessary [23]. Previously, we have studied the sensitivity of renal ultrasonography and renal scintigraphy in detecting grade 4–5 VUR, showing that both are useful in identifying the patients at greatest risk [21,22]. In the current study, we investigated whether both investigations could be used for grade 4–5 VUR risk scoring. Our results show that the sensitivity and specificity for detecting grade 4–5 VUR were 86% and 79%, respectively, in line with the corresponding values from our previous study [21]. It is noteworthy that DRF appears to be an independent risk factor for predicting grade 4–5 VUR.

In addressing VUR diagnosis across different age groups and clinical scenarios, guidelines from various urological associations provide a range of recommendations. The European Association of Urology advocates for the utilization of VCUG after the first f-UTI in infants and older children presenting with non-*E. coli* UTIs or recurrent infections, and in cases exhibiting upper tract dilatation in RBUS [33]. The British National Institute for Health and Care Excellence (NICE) similarly endorses VCUG for infants below six months of age experiencing atypical or recurrent UTIs, RBUS-detected dilatation, and non-*E. coli* infections, or in instances of familial VUR history [34]. Conversely, the American Urological Association (AUA) reserves VCUG for infants with SFU grade 3–4 hydronephrosis, hydroureter, abnormal RBUS findings, or those acquiring UTI during observation. Nevertheless, the AUA underscores the optionality of screening asymptomatic infants for VUR, citing a lack of evidential support from prospective studies [35]. Echoing a similar sentiment, The Urological Association of Asia suggests VCUG following repeated f-UTIs and when abnormalities are detected in RBUS or DMSA scans [36]. These varied guidelines highlight the intricate and nuanced aspects of VUR diagnosis, emphasizing the necessity for a personalized approach based on distinct clinical parameters. By utilizing our risk scoring system, we can concentrate on patients with higher scores, which are indicative of an elevated risk of high-grade VUR, and thereby reduce the number of patients requiring scanning. This approach not only improves diagnostic efficiency but also facilitates risk-stratified treatment planning tailored to each high-risk patient with antenatal hydronephrosis.

In our cohort, all patients initially presented with antenatal hydronephrosis rather than UTI, leading to the predominant use of MAG3/DTPA scintigraphy over DMSA for assessments on most of the patients. Given that MAG3/DTPA scintigraphy does not allow for the analysis of renal defects, we opted to use DRF, as it can be accurately assessed using both MAG3/DTPA and DMSA scintigraphy methods [37]. However, the impact of incorporating DRF into our scoring was modest, with the AUC being 0.89 when including DRF, compared to 0.85 when neglecting it. The slight improvement in the AUC, while potentially attributed to our limited sample size, represents an enhanced discriminative ability of the model through the inclusion of DRF, underscoring the potential utility of this parameter in risk stratification.

The results from our study align with those of others who have developed scoring systems for antenatal hydronephrosis. Chalmers et al. reviewed patients with antenatal hydronephrosis who had undergone both MRI and US, comparing grading systems and presenting a novel scoring system that improved inter-rater reliability, a known shortcoming in the SFU classification [38]. While their system enhances antenatal diagnosis, unlike ours, it does not reduce dependence on VCUG. Babu et al. devised a scoring system combining ultrasonographic and renographic parameters to differentiate pathological antenatal hydronephrosis, aiding in diagnostic streamlining [39]. This system showed good inter-rater reliability and is proposed to aid in deciding between operative and conservative treatment. Nelson et al. aimed to evaluate the inter-rater reliability of the consensus classification system for urinary tract dilatation components and overall scores in antenatal ultrasound examinations [40]. Compared to the SFU system, the UTD consensus fared better; however, agreement on parenchymal appearance still poses challenges. Dogan et al. used the UTD classification for VUR and scar detection and concluded that in patients with urinary tract infections and a UTD score of 0, a VCUG may not be necessary [41]. In contrast to our study, they focused on urinary tract infections to reduce VCUG dependency. Hodhod et al. introduced a scoring system based on MAG-3 renograms and RBUS, showing promise in predicting the need for operative intervention in a retrospective study [42]. Unlike our scoring system, theirs was devised for purposes other than streamlining differential diagnoses of antenatal hydronephrosis. A common issue in most antenatal hydronephrosis classification systems is the reliance on subjective interpretations of ultrasound findings, affecting reproducibility and reliability [43,44,45]. Additionally, the potential for reduced urinary output due to excessive postnatal weight loss may contribute to dehydration, raising the risk of neonatal false negative diagnoses [46]. In such instances, prenatal ultrasound findings have proven more reliable than postnatal assessments. This underscores the critical importance of vigilant postnatal urologic follow-up after prenatal indications, ensuring that early diagnostic insights are accurately corroborated and appropriately managed. Therefore, we focused on objective findings to avoid this limitation. However, further studies are still needed to assess the inter-observer reliability of our system.

Scoring systems for VUR are predominantly based on a history of urinary tract infections and are intended for a broader age range than ours, which aims to improve initial diagnostics before clinical symptoms emerge. Artificial intelligence shows promise in clinical medicine and could enhance diagnostic processes in the future [47]. However, further research is necessary before AI can be clinically applied to streamline the diagnostic paths of antenatal hydronephrosis [48,49,50]. Nevertheless, contemporary scoring systems based on AI exhibit potential and will no doubt improve future diagnostic processes through further refinement and iterative testing [51]. While the advent of AI and machine learning offers sophisticated methodologies for risk assessment, our scoring system, based on objective parameters, provides a straightforward, efficient, and reliable alternative that significantly reduces patient exposure to ionizing radiation. Further investigation with a larger data set is warranted to validate and possibly augment these findings. We believe that our scoring system is generally applicable across diverse healthcare systems at different levels and different resources, offering improved diagnostics for this vulnerable patient group.

The primary limitations of this study stem from its relatively small sample size and its retrospective design. Moreover, since the imaging studies were re-analyzed retrospectively, the measurements of bladder wall and renal cortex thickness were extracted from patient records, having been initially conducted by radiologists. Attempting to re-measure these dimensions from the stored ultrasonography images would have yielded unreliable results. However, a more comprehensive analysis that included these aspects might have revealed additional factors contributing to the risk classification, potentially enhancing its validity. This consideration highlights an area for future research, aiming to refine and improve the accuracy of our scoring system. Another constraint is that the DMSA scintigraphies were only performed on a minority of the patients; another limitation is that we had to use DRF instead of renal cortical defects in our analysis. These shortcomings not only underscore the limitations of our current study but also pave the way for future research. Highlighting key areas for further investigation, they promise advancements in our understanding and methodologies. Future efforts should focus on addressing these gaps and refining and enhancing this approach. This will ultimately strengthen the preemptive nature of interventions in antenatal hydronephrosis, leading to improved outcomes in this area. Nevertheless, to the best of our knowledge, this is the only study presenting data from infants with antenatal hydronephrosis, and the patients were carefully evaluated and followed up.

## 5. Conclusions

In patients with antenatal hydronephrosis, we combined previously identified indicators of significant risk for high-grade vesicoureteral reflux: a visible ureter and impaired unilateral renal growth in ultrasonography, coupled with reduced differential renal function in renal scintigraphy. Leveraging these parameters, we have developed a risk-scoring system based on ultrasonography and scintigraphy findings. This system stands out for its reliance on objective, quantifiable data, minimizing the potential for subjective interpretation and thereby ensuring high inter-rater reliability. As such, it is a valuable tool in guiding clinicians in patient selection for VCUG, enhancing diagnostic precision and patient care in the context of antenatal hydronephrosis.

## Figures and Tables

**Figure 1 diagnostics-14-00384-f001:**
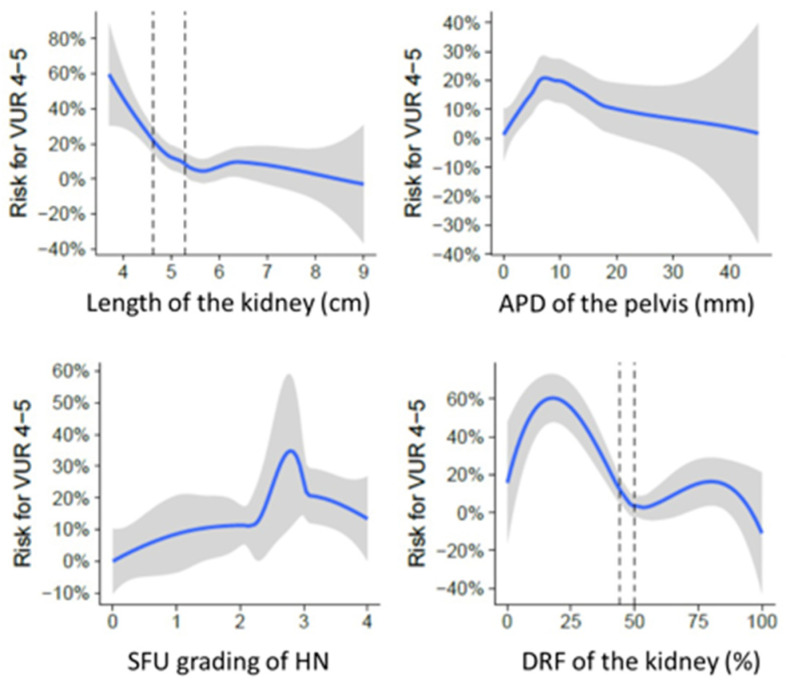
Risk for grade 4–5 VUR in relation to kidney length, anteroposterior diameter (APD) of the renal pelvis, SFU grading for hydronephrosis and differential renal function (DRF).

**Figure 2 diagnostics-14-00384-f002:**
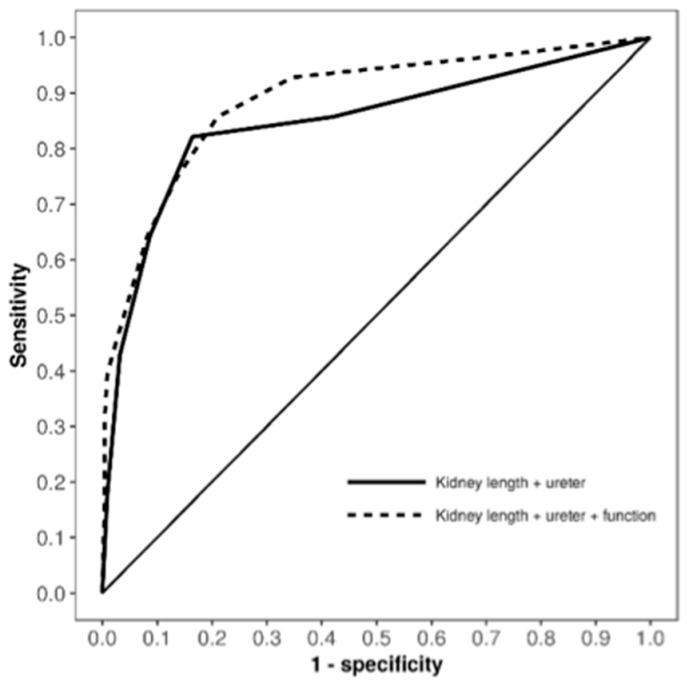
Receiver operating characteristic (ROC) curves for the risk factors (reduced renal length in standard deviation scale and visibility of the ureter in renal ultrasonography imaging and reduced differential renal function (DRF) in renal scintigraphy) for grade 4–5 VUR in patients with antenatally detected hydronephrosis.

**Table 1 diagnostics-14-00384-t001:** Patients with antenatal hydronephrosis, classified according to the higher grade of vesicoureteral reflux (VUR).

	Non-Refluxing Hydronephrosis	Grade 1–3 VUR	Grade 4–5 VUR
Males	69	6	18
Females	25	3	3
Total number of patients	94	9	21

**Table 2 diagnostics-14-00384-t002:** Scoring system for three-grade risk classification based on renal length, differential renal function (DRF) and a visible ureter.

Variable	Categories	Points	
Renal length	>53 mm	0	
	46–53	1	
	<46	2	
DRF	>50	0	
	44–50	1	
	<44	2	
Visible ureter	No	0	
	Yes	4	
**Risk group (controls/cases)**	**Score**	**Control (219)**	**Case (28)**
Low (79%/14%)	0	67	1
	1	76	1
	2	30	2
Intermediate (19%/43%)	3	16	3
	4	12	3
	5	14	6
High (2%/43%)	6	2	1
	7	1	2
	8	0	4

## Data Availability

Study data is available from authors on reasonable request.

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
