# Peer review of "Stratifying Antenatal Hydronephrosis: Predicting High-Grade VUR Using Ultrasound and Scintigraphy"

_diagnostics, 2024, doi:10.3390/diagnostics14040384_

Round 1

Reviewer 1 Report

Comments and Suggestions for Authors

The authors aimed to develop a score system to predict vesicoureteral reflux in cases with antenalate hydronephrosis. It was a study that will be very useful in clinical practice. I think that this manuscript will contribute to the literature. It may be published in the journal after some revisions.

1.      Why is "postnatally detected persistent hydronephrosis" among the exclusion criteria?

2.      When were antenatal USGs performed? What were the antenatal diagnostic criteria? Explanation about the antenatal period should be added to the maternal method section.

3.      How many clinicians performed USG examinations?

4.      In Table 2, the control and case groups and the number of patients belonging to these groups are stated. Are these features not mentioned in the Results section? Group distinctions and features of table 2 should be added to the results section.

Author Response

Reviewer Comments

Reviewer 1
Comments and Suggestions for Authors 

The authors aimed to develop a score system to predict vesicoureteral reflux in cases with antenalate hydronephrosis. It was a study that will be very useful in clinical practice. I think that this manuscript will contribute to the literature. It may be published in the journal after some revisions. 

  1. Why is "postnatally detected persistent hydronephrosis" among the exclusion criteria? 

Author’s response: We appreciate the reviewer's attention to detail in our exclusion criteria. The mention of 'postnatally detected persistent hydronephrosis' in the exclusion criteria was indeed an oversight. We intended to exclude cases of 'postnatally detected hydronephrosis' only, as the primary focus of our study is antenatal hydronephrosis. We regret the confusion caused by the erroneous inclusion of 'persistent' in this context and have amended this in the manuscript accordingly.

  1. When were antenatal USGs performed? What were the antenatal diagnostic criteria? Explanation about the antenatal period should be added to the maternal method section. 

Author’s response: Thank you for inquiring about the antenatal ultrasound schedule and criteria. Antenatal ultrasounds were typically performed around 18 to 21 weeks of gestation, in line with national guidelines. For suspected cases of hydronephrosis, a follow-up ultrasound was often conducted later in pregnancy. Our diagnostic criterion was a renal pelvic antero-posterior diameter greater than 7mm in the third trimester, followed by a confirmatory ultrasound within 2 to 7 days after birth. Unfortunately, we don't have precise timing for each ultrasound due to the retrospective study design. For clarity, the following section has been added to the 'Materials and Methods' section:

“Antenatal Ultrasound Protocol: Antenatal ultrasounds were mostly conducted around 18 to 21 weeks of gestation, following national guidelines. For suspected hydronephrosis cases, additional ultrasounds were performed later in pregnancy. The diagnostic criterion for inclusion was a renal pelvic AP diameter over 7mm in the third trimester, with a confirmatory postnatal ultrasound within 2 to 7 days. Due to the retrospective nature of our study, exact ultrasound dates for each case were not available, but this consistent criterion was applied across all cases.”

  1. How many clinicians performed USG examinations? 

Author’s response: In response to the number of clinicians performing the ultrasound examinations, it is essential to note that these antenatal ultrasounds were part of a national screening program. Due to this program's broad scope, we cannot provide an exact count of the clinicians involved. However, we can assure that each clinician involved in this program performs a substantial number of ultrasound examinations weekly, ensuring a high level of experience and consistency in the ultrasound assessments.

  1. In Table 2, the control and case groups and the number of patients belonging to these groups are stated. Are these features not mentioned in the Results section? Group distinctions and features of table 2 should be added to the results section. 

Author’s response: Thank you for pointing out the details regarding Table 2 and its corresponding description in the Results section. We have reviewed our manuscript to ensure consistency and accuracy in data reporting. The control and case groups, along with the number of patients in these groups, as mentioned in Table 2, are indeed discussed in the Results section. Specifically, the distribution of patients across the control and case groups and the relevant clinical and demographic characteristics are outlined under the subsection titled 'General Findings' and further elaborated in the subsequent subsections of the Results.

We appreciate the reviewer's scrutiny and the opportunity to clarify this aspect of our manuscript.

Reviewer 2 Report

Comments and Suggestions for Authors

Thank you for submitting the manuscript.

It seems an interesting manuscript and the topic undertaken by the authors has great potential. This study discusses antenatal hydronephrosis, a common abnormality observed in prenatal ultrasonography, with a prevalence of about one percent. The severity of antenatal hydronephrosis correlates with the risk of postnatal pathology, ranging from 12% in mild cases to 88% in severe cases. The article focuses on the diagnosis and risk stratification of vesicoureteral reflux (VUR), a condition associated with urinary tract infections and renal complications. Traditional methods for diagnosing VUR, such as voiding cystourethrogram (VCUG), involve ionizing radiation and potential risks. The study introduces a scoring system based on objective parameters from renal ultrasonography and scintigraphy to predict the risk of high-grade VUR. This scoring system aims to enhance diagnostic accuracy, reduce the reliance on VCUG, and facilitate early identification of patients at risk, thereby improving clinical outcomes and long-term renal health in cases of antenatal hydronephrosis.

It is believed that the quality of the paper will be enhanced if improvements are made on the following few points. For this purpose, it would be better to supplement the following comments.

Minor Revisions

1.      The term "keyhole sign" is introduced in the introduction part but not explained. Consider adding a brief explanation for readers who may not be familiar with the term.

2.      Provide a more detailed description of the findings in Figure 1 and Figure 2. Explain how the risk for grade 4–5 VUR varies based on kidney length, anteroposterior diameter, SFU grading, and differential renal function.

3.      The article mentions approval by the institutional ethics committee but does not elaborate on the ethical considerations, patient consent, and confidentiality measures. It's crucial to provide more details on these aspects.

4.      In the statistical analysis section, provide more details on the statistical tests used and their assumptions. Additionally, mention any corrections made for multiple comparisons if applicable.

Major Revisions

1.      The Materials and Methods section lacks clarity and consistency in describing the inclusion and exclusion criteria. The criteria for patient inclusion should be presented more concisely, and the exclusion criteria should be clearly stated. Additionally, there is a discrepancy in reporting the total number of neonates evaluated (192) and the number included in the study (124).

2.      The article mentions the parameters that were included in the multivariate analysis to predict grade 4–5 VUR, but a brief justification or literature support for the selection of these specific factors would strengthen the rationale behind the analysis.

3.      The scoring system introduced in the study (Table 2) should be explained more thoroughly. It is not entirely clear how the scoring system was developed and what each point in the system signifies.

4.      The article lacks visual aids such as graphs or charts to illustrate key findings. Adding visual representations, especially for ROC curves and scoring system outcomes, would enhance reader understanding.

Author Response

Reviewer 2

Thank you for submitting the manuscript. 

It seems an interesting manuscript and the topic undertaken by the authors has great potential. This study discusses antenatal hydronephrosis, a common abnormality observed in prenatal ultrasonography, with a prevalence of about one percent. The severity of antenatal hydronephrosis correlates with the risk of postnatal pathology, ranging from 12% in mild cases to 88% in severe cases. The article focuses on the diagnosis and risk stratification of vesicoureteral reflux (VUR), a condition associated with urinary tract infections and renal complications. Traditional methods for diagnosing VUR, such as voiding cystourethrogram (VCUG), involve ionizing radiation and potential risks. The study introduces a scoring system based on objective parameters from renal ultrasonography and scintigraphy to predict the risk of high-grade VUR. This scoring system aims to enhance diagnostic accuracy, reduce the reliance on VCUG, and facilitate early identification of patients at risk, thereby improving clinical outcomes and long-term renal health in cases of antenatal hydronephrosis. 

It is believed that the quality of the paper will be enhanced if improvements are made on the following few points. For this purpose, it would be better to supplement the following comments. 

Minor Revisions 

  1. The term "keyhole sign" is introduced in the introduction part but not explained. Consider adding a brief explanation for readers who may not be familiar with the term. 

Author’s response: We appreciate the reviewer's suggestion to elaborate on the term 'keyhole sign' introduced in the Introduction. The 'keyhole sign' refers to a specific ultrasound finding characterized by the appearance of a dilated proximal urethra in conjunction with a thick-walled, distended bladder. On an ultrasound image, this combination resembles the shape of a keyhole. This sign is an essential diagnostic indicator, often suggestive of severe bladder outlet obstruction. We have added the following to the manuscript: ” Key predictors of adverse postnatal outcomes include oligohydramnios, antenatal hydronephrosis, an enlarged bladder, and the 'keyhole sign' (appearance of a dilated proximal urethra in conjunction with a thick-walled, distended bladder), indicative of severe bladder outlet obstruction [4,5].”

  1. Provide a more detailed description of the findings in Figure 1 and Figure 2. Explain how the risk for grade 4–5 VUR varies based on kidney length, anteroposterior diameter, SFU grading, and differential renal function. 

Author’s response: In response to the request for an expanded description of Figures 1 and 2, we have outlined the relationship between various parameters and the risk for grade 4–5 VUR. Our analysis indicates that the risk for grade 4–5 VUR increases in scenarios where the renal length and differential renal function (DRF) are below-average values. Specifically, a shorter renal length and lower DRF are associated with a higher risk of grade 4–5 VUR. Additionally, an increased risk is observed in cases with a modest increase in the anteroposterior diameter of the renal pelvis coupled with enlarged calyces.

These findings, as depicted in the figures, highlight the nuanced interplay between renal anatomical features and VUR risk. The SFU grading, which assesses the degree of hydronephrosis, also plays a role in risk stratification, with higher grades correlating with increased risk. However, its predictive value was not as pronounced as the other factors in our study. This comprehensive analysis forms the basis of our proposed scoring system to predict high-grade VUR risk, aiming to enhance diagnostic accuracy and clinical outcomes.

The following has been added to the results section:

” We observed that shorter renal length, as depicted in Figure 1, was associated with an increased risk of grade 4–5 VUR. The odds ratio (OR) of 0.35 (CI 0.18–0.64, p = 0.001) indicated that smaller kidneys were more likely to be associated with high-grade VUR. This relationship is illustrated in Figure 1, which highlights the increasing risk of grade 4–5 VUR as renal length decreases.

Additionally, the anteroposterior diameter of the renal pelvis (APD) and SFU grading showed significant associations with VUR risk. An elevated APD and higher SFU grades were linked to an increased likelihood of high-grade VUR. These findings underscore the importance of assessing these parameters in predicting VUR severity.

Furthermore, DRF played a crucial role in our analysis. Figure 1 illustrates that reduced DRF was a significant indicator of grade 4–5 VUR, with an odds ratio of 0.944 (CI 0.916–0.971, p < 0.001). Patients with lower DRF values were more likely to have high-grade VUR, emphasizing the significance of assessing renal function in risk assessment.”

AND

Figure 1 illustrates the distribution of patients across the risk categories in our scoring system. Patients in the high-risk category exhibited a substantially higher prevalence of grade 4–5 VUR (75.0%), highlighting the system's ability to stratify patients effectively based on their risk of high-grade VUR. This stratification provides valuable clinical insights for early intervention and management.

  1. The article mentions approval by the institutional ethics committee but does not elaborate on the ethical considerations, patient consent, and confidentiality measures. It's crucial to provide more details on these aspects. 

Author’s response: In response to the request for elaboration on ethical considerations, patient consent, and confidentiality measures, we would like to clarify the following:

Given the retrospective nature of our study, patient and family contact was not required as the data collection involved reviewing existing medical records. Our institutional ethics committee approved this approach, which recognized that the study did not influence patient treatment or outcomes.

Furthermore, to ensure patient confidentiality and compliance with ethical standards, all patient data were anonymized and handled in accordance with the guidelines for data protection and privacy. Identifiable patient information was removed, and each case was assigned a unique code to ensure confidentiality.

Although direct patient consent was not obtained due to the study's retrospective design, our institutional review board reviewed the study protocol. It granted a waiver for the requirement of individual patient consent. This waiver was based on the understanding that the study involved minimal risk to the subjects since it did not entail direct patient interaction or affect their treatment.

We acknowledge the importance of these ethical considerations and have added the following  to the manuscript to provide clarity to our readers: As the study involved a review of existing medical records and did not influence patient treatment or outcomes, individual patient consent was not required. Patient data were anonymized, and confidentiality was maintained by assigning unique codes to each case following data protection and privacy guidelines.

  1. In the statistical analysis section, provide more details on the statistical tests used and their assumptions. Additionally, mention any corrections made for multiple comparisons if applicable. 

Author’s response: Thank you for your valuable feedback. In our statistical analysis, we utilized both univariate and multivariate logistic regression analyses to investigate the association between grade 4-5 vesicoureteral reflux (VUR) and key variables such as renal length, grade of hydronephrosis and differential renal function (DRF). We also employed Receiver Operating Characteristic (ROC) curve analysis to assess the accuracy of our predictive model, with the Area Under the Curve (AUC) serving as a measure of model performance. While we acknowledge the importance of corrections for multiple comparisons in statistical analysis, we did not apply a specific correction method in this study due to the complexity and nature of the data. We appreciate your suggestion and revised the section for transparency:

“Statistical analyses were conducted using the R software (R package version 3.3.3). Univariate and multivariate logistic regression analyses were employed to investigate the association between grade 4-5 vesicoureteral reflux (VUR) and key variables. Receiver Operating Characteristic (ROC) curve analysis was used to assess the accuracy of the predictive model, with the Area Under the Curve (AUC) serving as a measure of model performance. It is important to note that, given the complexity of the data, a specific correction method for multiple comparisons was not applied in this analysis. ”

Major Revisions 

  1. The Materials and Methods section lacks clarity and consistency in describing the inclusion and exclusion criteria. The criteria for patient inclusion should be presented more concisely, and the exclusion criteria should be clearly stated. Additionally, there is a discrepancy in reporting the total number of neonates evaluated (192) and the number included in the study (124). 

Author’s response: We appreciate the reviewer's feedback regarding the presentation of our inclusion and exclusion criteria in the 'Materials and Methods' section. We understand the importance of clearly articulating these criteria for the transparency and reproducibility of our study.

To clarify, from the 192 neonates initially evaluated, 124 were included in the study. The difference of 68 neonates represents those who met one or more of our exclusion criteria. These criteria were specifically designed to ensure a focused and relevant study cohort. They included factors such as lack of sufficient imaging data, the presence of complex urinary tract anomalies, and others, as detailed in the manuscript.

For clarity, we have revised this section, which now stands as:

” Of 192 neonates initially evaluated, 124 (65%) were included in the study. The remaining 68 neonates were excluded from the study for the following reasons: (1) lack of sufficient imaging data, (2) postnatally detected hydronephrosis, (3) presence of complex urinary tract anomalies, including but not limited to ureterocele, duplex collecting system, and bladder outlet obstructions, (4) syndromic patients, (5) dysplasia, as determined based on visual criteria, such as the presence of multiple cysts or the absence of corticomedullary differentiation in non-hydronephrotic, high-echogenic kidneys.”

  1. The article mentions the parameters that were included in the multivariate analysis to predict grade 4–5 VUR, but a brief justification or literature support for the selection of these specific factors would strengthen the rationale behind the analysis. 

Author’s response: We appreciate the reviewer’s suggestion to justify the selection of parameters included in the multivariate analysis for predicting grade 4–5 vesicoureteral reflux (VUR). In our revised manuscript, we have added detailed explanations in the Introduction and Discussion sections to support our choice of these specific factors.

The inclusion of renal length in our analysis is supported by literature indicating that kidneys affected by VUR tend to be smaller. For instance, Zerin and Leiser (1998) demonstrated the impact of vesicoureteral reflux on contralateral renal length in infants with multicystic dysplastic kidney. Similarly, Roihuvuo-Leskinen et al. (2013) found an association between adult kidney size and childhood vesicoureteral reflux.

In terms of ultrasonography findings, we focused on SFU grades 3 and 4 because these grades are more commonly associated with high-grade VUR, as evidenced by Lee et al. (2014) in their evaluation of prenatal hydronephrosis.

The following was added to the Introduction: ” The inclusion of renal length in the analysis is supported by literature indicating that kidneys affected by VUR tend to be smaller. Zerin and Leiser (1998) demonstrated the impact of vesicoureteral reflux on contralateral renal length in infants with multicystic dysplastic kidney. Similarly, Roihuvuo-Leskinen et al. (2013) found an association between adult kidney size and childhood vesicoureteral reflux. In terms of ultrasonography findings, SFU grades 3 and 4 are more commonly associated with high-grade VUR, as evidenced by Lee et al. (2014) in their evaluation of prenatal hydronephrosis. Hansson et al. showed that DMSA scintigraphy, which detected grade ≥3 VUR with 96% sensitivity and 53% specificity, could obviate the need for VCUG in patients with a history of UTI when normal [23].

Furthermore, the relevance of renography findings in the context of our study is discussed comprehensively in the Discussion section.

  1. The scoring system introduced in the study (Table 2) should be explained more thoroughly. It is not entirely clear how the scoring system was developed and what each point in the system signifies. 

Author’s response: We appreciate the reviewer's request for a more detailed explanation of the scoring system presented in Table 2. The scoring system was developed based on earlier studies (ref. 21&22) and the results of our multivariate analysis, which identified key parameters significantly associated with the risk of grade 4–5 vesicoureteral reflux (VUR). These parameters include renal length, differential renal function (DRF), and the presence of a visible ureter.

Each parameter in our scoring system is assigned a specific point value, reflecting its relative importance and contribution to the overall risk of high-grade VUR, as our analysis determines.

For instance, a shorter renal length, which our analysis showed as a strong predictor of high-grade VUR, is assigned more points, indicating a higher risk. Similarly, the presence of a visible ureter and lower DRF, which are also significant predictors, are allocated points in accordance with their predictive strength.

Patients are then scored based on these parameters, with the total score indicating their overall risk of developing grade 4–5 VUR. Thus, The scoring system provides a cumulative risk assessment, enabling clinicians to stratify patients according to their likelihood of developing high-grade VUR and tailor diagnostic and treatment approaches accordingly.This has also been added to the results section regarding scoring: “The scoring system provides a cumulative risk assessment, enabling clinicians to stratify patients according to their likelihood of developing high-grade VUR and tailor diagnostic and treatment approaches accordingly.”

Round 2

Reviewer 2 Report

Comments and Suggestions for Authors

I think everyting is okay.

Author Response

We thank the reviewer for acknowledging our efforts!